# Hepatitis C prevalence in Denmark in 2016—An updated estimate using multiple national registers

Stine Nielsen[1], Janne Fuglsang Hansen[1], Gordon Hay[2], Susan Cowan[3], Peter Jepsen[4,5], Lars Haukali Omland[6], Henrik Bygum Krarup[7,8], Jacob Søholm[1], Jeffrey V. Lazarus[9], Nina Weis[10,11], Anne Øvrehus[1,12], Peer Brehm Christensen[1,12]*

1 Department of Infectious Diseases, Odense University Hospital, Odense, Denmark, 2 Centre for Public Health, Liverpool John Moores University, Liverpool, United Kingdom, 3 Department of Infectious Disease Epidemiology and Prevention, Statens Serum Institut, Copenhagen, Denmark, 4 Department of Clinical Epidemiology, Aarhus University Hospital, Aarhus, Denmark, 5 Department of Hepatology and Gastroenterology, Aarhus University Hospital, Aarhus, Denmark, 6 Department of Infectious Diseases, Copenhagen University Hospital, Rigshospitalet, Denmark, 7 Department of Molecular Diagnostics, Aalborg University Hospital, Aalborg, Denmark, 8 Department of Medical Gastroenterology, Aalborg University Hospital, Aalborg, Denmark, 9 Barcelona Institute for Global Health (ISGlobal), Hospital Clínic, University of Barcelona, Barcelona, Spain, 10 Department of Infectious Diseases, Copenhagen University Hospital, Hvidovre, Denmark, 11 Department of Clinical Medicine, Faculty of Health and Medical Sciences, University of Copenhagen, Copenhagen, Denmark, 12 Department of Clinical Research, Faculty of Health Sciences, University of Southern Denmark, Odense, Denmark

* Peer.christensen@dadlnet.dk

**Data Availability Statement:** The dataset for the manuscript has been published at Zenodo.org: DOI 10.5281/zenodo.3959476.

## Abstract

### Background

Chronic hepatitis C (CHC) can be eliminated as a public health threat by meeting the WHO targets: 90% of patients diagnosed and 80% treated by 2030. To achieve and monitor progress towards elimination, an updated estimate of the size of the CHC population is needed, but Denmark has no complete national CHC register. By combining existing registers in 2007, we estimated the population living with CHC to be 16,888 (0.38% of the adult population).

### Aim

To estimate the population living with diagnosed and undiagnosed CHC in Denmark on 31 December 2016. Among additional aims were to estimate the proportion of patients attending specialised clinical care.

### Methods

People with diagnosed CHC were identified from four national registers. The total diagnosed population was estimated by capture-recapture analysis. The undiagnosed population was estimated by comparing the register data with data from two cross-sectional surveys.

### Results

The population living with diagnosed CHC in Denmark was 7,581 persons (95%CI: 7,416–12,661) of which 6,116 (81%) were identified in the four registers. The estimated undiagnosed

**Funding:** The study was supported by an unrestricted grant from MSD Denmark URL: https://www.msd.dk/home. The funders had no role in study design, data collection and analysis, decision to publish, or preparation of the manuscript.

**Competing interests:** Names of funding sources: MSD Denmark, (Merck Sharp & Dohme) The funder had no role in the study design; collection, analysis, and interpretation of data; writing of the paper; and/or decision to submit for publication None of the authors has served or currently serve on the editorial board of PlosOne None of the authors has acted as an expert witness in relevant legal proceedings None of the authors has sat or currently sit on a committee for an organization that may benefit from publication of the paper The funding from MSD was used for dataextraction, datacleaning, and statistical expertise during the analysis of data. This does not alter our adherence to PLOS ONE policies on sharing data and materials.

fraction was 24%, so the total CHC infected population was 9,975 corresponding to 0.21% of the adult population (95%CI: 9,758–16,659; 0.21%-0.36%). Only 48% of diagnosed patients had received specialised clinical care.

## Conclusion

CHC prevalence in Denmark is declining and 76% of patients have been diagnosed. Linking diagnosed patients to care and increasing efforts to test people with former or current drug use will be necessary to achieve CHC elimination.

## Introduction

Chronic hepatitis C (CHC) is a major health problem worldwide, but new treatments have made elimination possible [1, 2]. The World Health Organization (WHO) has set the ambitious targets of 90% diagnosed and 80% treated by 2030, which was endorsed by all Member States, including Denmark, in 2016 [3]. In the same year, Denmark also endorsed the action plan for the health sector response to viral hepatitis in the WHO European Region [4].

Achieving the elimination goal requires knowing the size of the population living with CHC. This is also a cornerstone in any national elimination plan. However, unlike many countries in the world, Denmark has not formulated a national viral hepatitis strategy or action plan.

Denmark is well known for its high coverage national health registers and in 2012, we published the first Danish CHC estimate based on four national registers using a capture-recapture model [5]. This method gives an estimate of a 'hypothetical' population which would have been diagnosed if one or more of the registers included 100% of all diagnosed individuals [6]. We calculated a diagnosed population of 9,166 and estimated the total population (diagnosed and undiagnosed) living with CHC to be 16,888 (0.38% of the adult population) at the end of 2007. However, due to incomplete reporting to the registers, the total CHC population could have been as high as 21,468 (0.49%). An update is now urgently needed to plan the CHC elimination efforts in Denmark. The primary aim of this study was to estimate the population with diagnosed and undiagnosed CHC in Denmark at the end of 2016. Secondary aims were to estimate the proportion of patients attending specialised clinical care and the coverage of the national CHC registers.

## Methods

### Data sources

We used the same four national source registers as in our previous estimate from 2007 [5].

1. Communicable diseases register: Since May 2000, Denmark has implemented mandatory reporting of CHC from the diagnosing physician. The CHC case definition is based on the presence of hepatitis C virus RNA (HCV-RNA). This register is known to have low (<50%) coverage [7].

2. Hospital register: In 1977, Denmark established the Danish National Patient Registry, which included all inpatient discharge diagnoses, according to ICD-8 and ICD-10. Since 1995, it also included all hospital outpatient and emergency department visits [8]. We extracted data on all individuals registered with CHC (ICD-10 diagnosis B18.2). The case

definition for CHC is not specified. The validity of the register was approximately 67% for gastroenterology disease diagnoses in 2002 [9]. A 2004 study found that 48% of CHC/HIV co-infected patients were recorded with a CHC diagnosis in the hospital register [10].

3. Clinical hepatitis database (DANHEP): In 2002, Denmark established the Danish Database for Hepatitis B and C. It contains detailed information on demographics, test results, treatment status, risk factors, co-morbidities and other relevant information on all chronic hepatitis B and CHC patients seen for care in specialised clinics in Denmark. Patients positive for HCV-RNA in their most recent test were classified as having CHC.

4. Laboratory database (Danvir): Denmark has 18 laboratories performing tests for hepatitis C [5]. We requested information on all people ever tested for either hepatitis C (antiHCV or HCV-RNA test) or hepatitis B. The case definition for CHC was positive HCV-RNA at last test. People with negative HCV-RNA at last test or those only antiHCV positive were excluded. In contrast to the other three source registers, the laboratory register was not updated automatically; it is only updated on request from the research team, and not all laboratories provided fully updated data.

For patients present in the source registers, we extracted data from two additional registers: The Danish civil register: established in 1968 and stores information on vital status, current place of residence as well as immigration/emigration on all Danish residents [11].

The drug treatment register: established in 1996 and contains information on all persons treated for drug use in Denmark [12]. This register was used to explore how many people with diagnosed CHC had been in contact with drug treatment services.

All persons with permanent residence in Denmark are assigned a unique 10-digit personal identification number (PIN). This number was used to link the information on individuals from the different registries. The total population in Denmark at the end of 2016 was 5.7 million, of which the adult population (≥18 years) was 4.6 million (80%) [13].

We identified the individuals who fulfilled the CHC case definition in each register and excluded those who had died, left Denmark or had an invalid PIN as of 31 December 2016.

We excluded those who had cleared their hepatitis C virus (HCV) infection (i.e. their last HCV-RNA test was negative and/or they were classified as cured in the clinical database). Information about cleared infections was only available from the laboratory register and the clinical database. Therefore, we also excluded patients from the hospital and communicable disease register if their last registration in these registers were before the date of CHC clearance in the other registers. We removed patients who were only registered with CHC in the hospital register but were "non-CHC cases" according to other registers (e.g. they were HCV-RNA negative or not tested for HCV-RNA and either antiHCV negative or hepatitis B virus positive in any of the other registers).

The following variables were extracted for all cases as of 31 December 2016: First year of diagnosis (i.e. first year the person appeared in any of the four registers), age, sex, region of current residence and treatment for drug use.

## Capture-recapture estimation

Assuming independence between the source registers and a common CHC case definition, we analysed the overlap patterns between the four registers stratified by age (3 groups), sex (2 groups), geographic region (5 groups) and first year of diagnosis (3 groups). As some cells had too few observations to produce valid estimates, these were analysed without year of diagnosis to obtain stable estimates. We carried out log-linear modelling using the statistical program GLIM4 [14] and used the same analytical approach as in 2007 [5, 15]. The final analysis

contained 113 different models including all possible two-way and three-way interactions fitted to the overlap data. Confidence intervals for the total estimate were derived from bootstrap analysis of 1000 samples [16].

### Estimating the undiagnosed fraction

We used data from two cross-sectional studies to estimate the proportion of undiagnosed CHC in Denmark:

1. A regional study of older adults (The "3B-study"): Retrospective testing of 4,945 stored blood samples originally collected between 1998–2000 to investigate Helicobacter pylori infection [17, 18]. The population tested in this study was 58–83 years old in 2016. The stored samples were tested for HCV-RNA in 2014.

2. A seroprevalence study of HCV infection among 1,041 people in prison conducted between 2016–2017 where 801 (77%) were tested for HCV-RNA. The study was performed in eight prisons in the South Region. Participants came from all over Denmark and were representative of the national prison population. The median age of participants was 30 years, 97% were male, and 8.5% reported injecting drug use [19].

We investigated if individuals identified with CHC in these two studies were diagnosed with CHC in any of the source registers and calculated a 95% confidence interval on the estimated undiagnosed fraction.

### Sensitivity analyses

To investigate the impact that under-reporting and misclassification of hepatitis cases might have on our estimate, we performed several sensitivity analyses. We performed three source capture-recapture analyses to evaluate the effect of excluding one of the four source registers [20].

In addition, we used a multiple indicator method (MIM) analysis to validate an unrealistically high estimate for the South Region in the capture-recapture analysis [21]. In this multiple regression model, the CHC prevalence (per capita) was the dependent variable and the 'independent' variables were the four source registers. The model with only the hospital register had the best fit across regions, and this was not improved by including other source registers.

We followed the WHO guidelines for accurate and transparent health estimates reporting [22]. The study was approved by the Danish Data Protection Agency in 2016 (Journal no: 16/43190 and 18/52996). Data were fully anonymized prior to analysis. As a register-based study without contact to patients, informed consent from participants was not required, according to Danish law. However patients registered in the clinical database (DANHEP) did provide written consent to have their data used in research prior to registration in the database.

### Results

The initial extraction included 1,046,013 individuals, of whom 20,174 were identified as ever registered as infected with hepatitis C. Of these, 6,380 (32%) had died and 1,188 (6%) did not have a valid PIN, resulting in 12,606 individuals (62%) being included for further analysis. Among these, 9,973 (79%) people had ever had a positive HCV-RNA test, including 2,565 (26%) who had cleared their infection (of which 1,604 (63%) had recorded successful CHC treatment). Excluding patients with past infection left 7408 (74%) patients for further analysis.

Among the 7408 patients the overlap between the four registers varied: most cases were in all four registers and the second-largest group was those, only present in the hospital register.

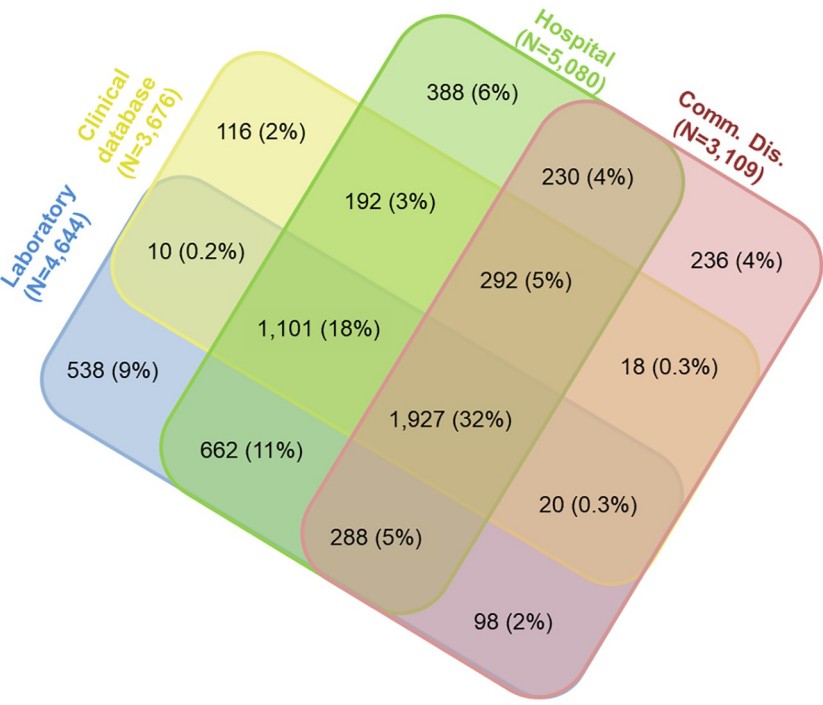

**Fig 1. Overlap pattern between chronic hepatitis C cases in each of the four Danish registers (N = 6,116) after validation.**

Of 1,681 patients, only in the hospital register, we identified 1,292 who had either negative antiHCV or HCV-RNA results and/or had chronic hepatitis B (ICD10 code 18.1) with no signs of HCV co-infection according to the other source registers. We could not check the clinical records of these patients, but in the "3B-study" with complete laboratory data, 7 patients only diagnosed in the hospital register were all found to be non-CHC cases. We excluded the 1,292 (17%) non-CHC patients reducing the "Hospital only" group to 388 and our total captured population to 6,116 (Fig 1).

Among the 6,116 living with CHC, most were identified through the hospital register (N = 5,080, 83%) and the laboratory database (N = 4,644, 76%) and 60% had attended specialised clinical care (Table 1).

The median age was 50 years (with 80% being between 36–64 years) and 65% were male. Most living with diagnosed CHC were aged between 50–59 years in 2016 (N = 1,938, 32%). The age at diagnosis had increased with time: the median age of those with first entry in any register before 2001 was 33 years and increased to 44 years for those diagnosed after 2008. The majority of cases were among people living in the Capital region (37%) and overall, most people (38%) were diagnosed after 2008. More than half (58%) of those with diagnosed CHC had ever been in treatment for drug use, ranging between 54% in the Capital and 65% in the South Region.

## Capture-recapture results

The capture-recapture analysis suggests that if the registers were not subject to under-reporting there would be, in total, 7,581 people (95%CI 7,416–12,661) living with diagnosed CHC in Denmark. This represents a 17% decrease over 9 years (N = 9,166 diagnosed cases in 2007 [5]).

**Table 1. People living with diagnosed chronic hepatitis C in Denmark according to four national registers end 2016 (N = 6,116).**

| Database | Laboratory | | Comm. Dis | | Hospital | | Clinical | | TOTAL | |
|---|---|---|---|---|---|---|---|---|---|---|
| **Chronic hepatitis C** | **4644** | 76% | **3109** | 51% | **5080** | 83% | **3676** | 60% | **6116** | 100% |
| only in one register | 538 | 9% | 236 | 4% | 388 | 6% | 116 | 2% | 1278 | 21% |
| **Sex** | | | | | | | | | | |
| Male | 3069 | 66% | 2009 | 65% | 3278 | 65% | 2341 | 64% | 3986 | 65% |
| **Age** | | | | | | | | | | |
| <40 | 700 | 15% | 516 | 17% | 871 | 17% | 682 | 19% | 1052 | 17% |
| 40–49 | 1412 | 30% | 990 | 32% | 1451 | 29% | 1067 | 29% | 1795 | 29% |
| 50+ | 2532 | 55% | 1603 | 52% | 2758 | 54% | 1927 | 52% | 3269 | 53% |
| **Administrative region** | | | | | | | | | | |
| North | 390 | 8% | 168 | 5% | 397 | 8% | 317 | 9% | 490 | 8% |
| Central | 782 | 17% | 392 | 13% | 833 | 16% | 680 | 18% | 962 | 16% |
| South | 1243 | 27% | 910 | 29% | 1183 | 23% | 1002 | 27% | 1595 | 26% |
| Zealand | 524 | 11% | 480 | 15% | 688 | 14% | 369 | 10% | 825 | 13% |
| Capital region | 1705 | 37% | 1159 | 37% | 1979 | 39% | 1308 | 36% | 2244 | 37% |
| **Year of diagnosis** | | | | | | | | | | |
| ≤2000 | 1462 | 31% | 1000 | 32% | 1487 | 29% | 999 | 27% | 1787 | 29% |
| 2001–2007 | 1743 | 38% | 1104 | 36% | 1617 | 32% | 1248 | 34% | 1990 | 33% |
| 2008–2016 | 1439 | 31% | 1005 | 32% | 1976 | 39% | 1429 | 39% | 2339 | 38% |
| **Registered in the drug treatment register** | 2804 | 60% | 2014 | 65% | 2917 | 58% | 2119 | 58% | 3556 | 58% |

The total "hidden" diagnosed population was 1,465 (19%) but varied between 6% in the North and 31% in the South Region (Table 2).

## Estimating the fraction with undiagnosed CHC

We used two cross-sectional studies where, in total, N = 12 (the "3B-study") and N = 34 (prison study) participants respectively tested HCV-RNA positive. Of these, 4 (33%) and 7 (21%) were not found in any of the four source registers and we therefore classified in total 24% (11/46) (95% CI 13%-39%) to have undiagnosed CHC.

Applying this to our estimated 7,581 diagnosed chronic infections resulted in a total estimated population living with CHC (diagnosed plus undiagnosed) of 9,975 (95% CI 9,758–16,659). This corresponded to a national adult CHC prevalence of 0.22% (95% CI 0.21%-0.36%), significantly lower than the 0.38% reported in 2007 (p<0.05) [5].

## Sensitivity analyses

We addressed several issues in our data:

Firstly, if we had not excluded the 1,292 misclassified CHC cases, the estimated diagnosed population would be 11,158 (95%CI 10,489–15,630).

Secondly, the major decline in reporting from laboratories after 2010, not mirrored in the other registers reflects that the laboratory database was not automatically updated (Fig 2).

If reports from the laboratory had continued to follow the hospital register after 2010, then approximately 700 additional cases would have been reported. A simple three-source capture-recapture estimate [20] excluding the laboratory register and with no correction for interactions increased the estimated diagnosed population by 24% (N = 1,850), whereas excluding the hospital register meant an increase of 9% (N = 669).

Thirdly, the South Region had the highest proportion of diagnosed patients not present in the registers (the hidden population). This was unexpected as this region has been a pioneer in

**Table 2. Estimated number of people living with chronic hepatitis C (diagnosed and undiagnosed) in Denmark at the end of 2016 (N = 9,975).**

| | North | Central | South | Zealand | Capital | Denmark | 95% CI |
|---|---|---|---|---|---|---|---|
| | n (%) | n (%) | n (%) | n (%) | n (%) | n (%) | |
| **Age** | | | | | | | |
| <40 | 100 (19) | 193 (17) | 531 (23) | 146 (15) | 365 (14) | 1,335 (18) | 1,199–2,996 |
| 40–49 | 136 (26) | 329 (29) | 811 (35) | 267 (27) | 707 (27) | 2,250 (30) | 2,088–3,663 |
| 50+ | 283 (55) | 628 (55) | 959 (42) | 561 (58) | 1,565 (59) | 3,996 (53) | 3,815–7,199 |
| **Sex** | | | | | | | |
| Male | 342 (66) | 756 (66) | 1,613 (70) | 592 (61) | 1,579 (60) | 4,882 (64) | 4,742–8,127 |
| **Year of diagnosis** | | | | | | | |
| ≤ 2000 | 195 (38) | 356 (31) | 728 (32) | 249 (26) | 569 (22) | 2,097 (28) | 1,954–2,290 |
| 2001–2007 | 141 (27) | 392 (34) | 714 (31) | 295 (30) | 989 (38) | 2,531(33) | 2,344–6,316 |
| 2008–2016 | 183 (35) | 402 (35) | 859 (37) | 430 (44) | 1,079 (41) | 2,953 (39) | 2,789–4,994 |
| **Total estimated diagnoses** | 519 (7) | 1,150 (15) | 2,301 (30) | 974 (13) | 2,637 (35) | **7,581 (100)** | **7,416–12,661** |
| 95% CI | 507–619 | 1,052–2,397 | 2,044–5,141 | 929–1,212 | 2,465–5,232 | | |
| **"Hidden" CHC diagnoses**[a] | 29 (6) | 188 (16) | 706 (31) | 149 (15) | 393 (15) | 1,465 (19) | 1,300–6,545 |
| **Total CHC population**[b] | **683** | **1,513** | **3,028** | **1,282** | **3,470** | **9,975** | **9,758–16,659** |
| **Population prevalence**[c] | 0.14% | 0.15% | 0.31% | 0.19% | 0.24% | 0.22% | 0.21%-0.36% |

[a] The "hidden" CHC diagnoses is the main outcome of the capture-recapture analyses and refers to the estimated number of CHC diagnoses not identified due to incompleteness of the registers (i.e. it is the total estimated diagnoses minus the observed ("captured") diagnoses).

[b] Total CHC population: adjustment for 24% undiagnosed.

[c] Population prevalence: estimated CHC prevalence (diagnosed and undiagnosed) in the adult population.

outreach hepatitis C testing for decades. Completeness of registers in the South Region is believed to be high, and "laboratory only" was 14% here compared to 9% in the national estimate (Fig 1). The capture-recapture analyses in the South Region tended to favour more complex models (with more interaction between source registers than the other areas). These factors could lead to over-estimation. To address this, we adjusted CHC prevalence in the South using a multiple indicator method (MIM) regression model. This reduced the estimated diagnosed CHC population in the South from 2,301 to 1,599 (95% CI 1,308–1,890) compared to 1,595 observed cases, suggesting only 0.2% hidden in the region, an unlikely high reporting

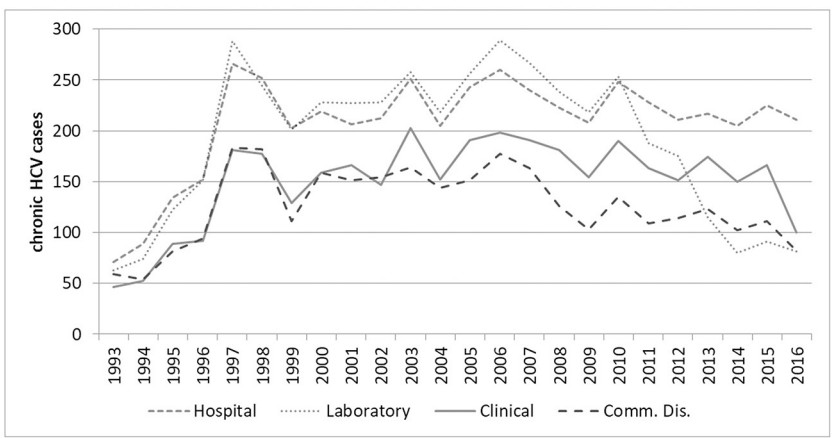

**Fig 2. Number of new chronic hepatitis C cases reported per year in the four source registers (N = 6,116).**

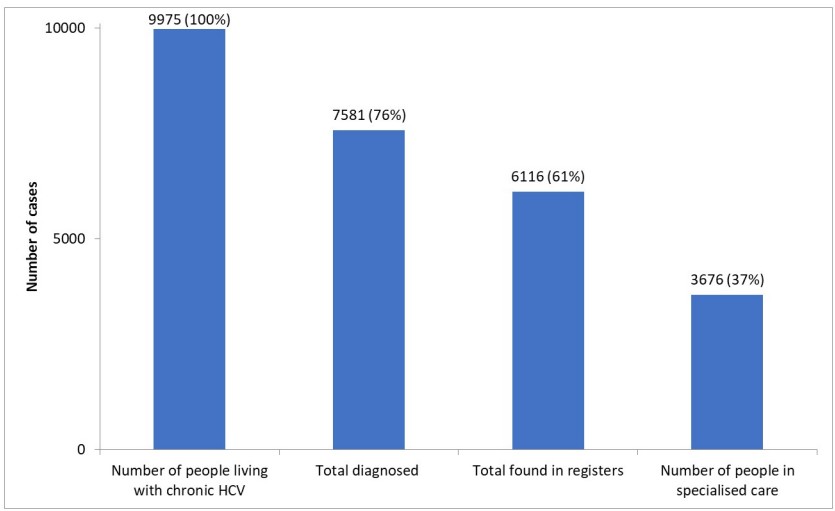

**Fig 3. The estimated continuum of care for chronic hepatitis C in Denmark (end 2016).**

rate. If we accept the MIM adjustment for the South, the national estimate would decrease by 9% to 6,879 (95% CI 6,674–10,215).

Fourthly, antiHCV positive patients without a positive HCV-RNA result were excluded. This reflected that previously, HCV-RNA was only ordered by hospital specialists once the patient was referred, thus patients tested in drug treatment facilities and primary care, but not entering specialised care, did not get HCV-RNA tested. In the 2007 study, 62% of antiHCV positive patients tested with PCR were HCV-RNA positive [5]. Thus by excluding the antiHCV only patients we underestimated the chronically infected population.

In summary, the sensitivity analysis indicated both under- and over-estimations, and the point estimate of 9,975 is likely to be a conservative estimate given the limitations of our source registers.

Combining our results in a national cascade of care showed that 37% of all people living with CHC in Denmark had attended specialised clinical care (Fig 3).

## Discussion

This study estimated the population living with CHC in Denmark to be 9,975 (95% CI 9,758–16,659) including an undiagnosed fraction of 24%. From 2007 to 2016, the estimated CHC prevalence fell from 0.38 to 0.21%, and the diagnosed fraction increased from 54% to 76%. However, comparison is difficult as we did not exclude patients misclassified in the hospital register in the 2007 estimate. Still we think that using the same registers and capture recapture methodology makes the observed decline in prevalence over time credible.

One reason for the declining prevalence was a high mortality in the cohort; 32% of all in the registers exposed to hepatitis C had died. In addition, 10% had been cured of CHC. Another force driving the reduction in CHC prevalence in Denmark was a low CHC prevalence in young people who use drugs [23]. Moreover, several studies suggest that fewer young people inject drugs in later years [12, 19, 23]. The very low CHC prevalence among those younger than 40 years also suggests a low incidence of new infections. In contrast, the prevalence among 50+ years has doubled the last 10 years, reflecting an ageing cohort effect.

Our findings are in line with a recent study from England, which estimated the CHC prevalence in 2015 to be 0.27% in the adult population and a 10-year decrease of 23% from 2005,

compared to our decrease of 42% from 2007 (0.38%) to 2016 (0.21%) [24]. However, estimates in England suggest that only 31% of people living with CHC have been diagnosed. Our estimate of the diagnosed proportion (76%) was based on two very small samples ranging between 67–79%, of which only one (the prison study) had national coverage. This is similar to the findings from a recent modelling study which estimated that in 2015 63% of CHC cases in Denmark were diagnosed [25]. The general population sample was older than our register cohort and the prison study participants were younger. Applying the same proportion of undiagnosed CHC in all regions assumes that the testing coverage and reporting was similar between regions, but this is unlikely. It is expected that higher test coverage would be found in persons born 1950–1980 as these birth-cohorts have been the focus for testing in Denmark inspired by the "baby boomer" testing initiative in the US. On the other hand, the 3B-study was performed in Funen, a region where the focus on HCV testing has been high, and it is possible that this is lower in other regions. Our diagnosed proportion was similar to Sweden, where a recent hepatitis C survey found that 73% of people who tested antiHCV positive had been previously diagnosed [26]. A recent systematic review found that 13 EU/EEA countries had conducted HCV prevalence surveys with an estimated antiHCV prevalence in the EU/EEA of 1.1% (95% CI 0.9–1.4%) of which an estimated 70% have CHC [27]. In the Netherlands, a CHC prevalence of 0.16% was found using the workbook method, whereas Ireland and Belgium reported 0.98% and 0.13% CHC prevalence using residual sera testing [28–30]. However, none of these methods were directly comparable to our study.

We were surprised to see the variation in prevalence between the different regions, and, especially the higher prevalence (and larger diagnosed population not in the registers) in the Southern Region than in the Capital Region. CHC is strongly associated with injecting drug use and half of the people using drugs in Denmark are believed to live in Copenhagen, so we would expect a much higher prevalence here than observed. As the South Region had more focus on testing people who use drugs for HCV, the difference could be explained by more people tested in the South, where registers were updated until end 2016. However, this would imply a lower hidden population in the South, in contrast to what our model predicted. This was also suggested by the MIM analysis, although this resulted in an unrealistically high diagnosed proportion in the South Region (99.8%). So all things considered, we believe that the hidden population in the Southern Region is likely over-estimated.

The coverage of the national communicable disease register had risen from 32% in 2007 to 41% in 2016. This still relatively low coverage reflects that the register relies on clinicians reporting as Denmark is one of few European countries without mandatory laboratory reporting for hepatitis C [31].

The proportion of diagnosed patients attending clinical care had risen from 34% (3,065/9,166) in 2007 to now 48% (3,676/7,581), but still more than half of those living with diagnosed CHC are not attending specialised care. This suggests that calling in patients not attending care and offering them treatment could be more cost-effective than increasing screening for the remaining undiagnosed CHC patients in a low prevalence population like Denmark.

There are a number of weaknesses in our study. Firstly, the basic assumption of independence of registers in the capture-recapture analysis was not fulfilled. We used log-linear modelling including interaction terms between registers to compensate for this, but with only limited success. Secondly, the case definition was not the same in all registers. Particularly, in the hospital register, the case definition was probably not always based on HCV-RNA positivity: patients are usually coded by the discharging doctor or by administrative staff and HCV-RNA results may not be available at the time of discharge.

Differences in case definitions will decrease the overlap between registers and inflate the estimate. We tried to adjust for this by applying another statistical method (MIM), but this resulted in too low an estimate for the South Region.

Our estimate of the proportion with undiagnosed infection was based on small numbers. This was because the CHC prevalence was low and general population serological surveys, which enable assessment of the undiagnosed fraction, are rarely conducted in Denmark. Also, recent sentinel studies in key populations, like people who inject drugs or migrants from high prevalence areas, were not available.

A major weakness was that the laboratory register we used was a research database with incomplete reporting from the participating laboratories. Efforts are currently underway to implement mandatory laboratory-based reporting of CHC in Denmark, which would simplify future assessments of the national hepatitis C burden. Finally, we did not consider reinfection among the cured. However, with only 1,604 successfully treated, this would probably only add few extra cases.

## Conclusion

Our study suggests that the population living with CHC in Denmark in 2016 was 9,975 (9,758–16,659) people and declining. Only 41% had been reported to the national communicable diseases register and less than half had attended specialised care. The relatively large range on our estimate, highlights the methodological challenges and uncertainties. However, no better evidence-based estimate exists and we believe this result can assist the health authorities to formulate a national elimination plan that can assure that Denmark will fulfil the WHO hepatitis C elimination goal by 2030. According to our study, the most urgent initiative will be to ensure that infected people are linked to care and that treatment is offered to all diagnosed patients, and this strategy has recently been accepted by the national health authorities (PBC personal communication). People with current or former drug use constitute the main CHC population in Denmark and thus addressing the needs of this population will be key to reach national elimination.

## Acknowledgments

Anderson Rael dos Santos (Liverpool John Moores University) carried out the capture-recapture analyses.

## Author Contributions

**Conceptualization:** Stine Nielsen, Peer Brehm Christensen.

**Data curation:** Stine Nielsen, Janne Fuglsang Hansen, Susan Cowan, Peter Jepsen, Lars Haukali Omland, Henrik Bygum Krarup, Nina Weis, Peer Brehm Christensen.

**Formal analysis:** Stine Nielsen, Janne Fuglsang Hansen, Gordon Hay, Jacob Søholm, Anne Øvrehus, Peer Brehm Christensen.

**Funding acquisition:** Jeffrey V. Lazarus, Peer Brehm Christensen.

**Methodology:** Stine Nielsen, Gordon Hay, Jeffrey V. Lazarus.

**Project administration:** Peer Brehm Christensen.

**Supervision:** Peer Brehm Christensen.

**Validation:** Stine Nielsen, Janne Fuglsang Hansen.

**Writing – original draft:** Stine Nielsen, Peer Brehm Christensen.

**Writing – review & editing:** Stine Nielsen, Janne Fuglsang Hansen, Gordon Hay, Susan Cowan, Peter Jepsen, Lars Haukali Omland, Henrik Bygum Krarup, Jacob Søholm, Jeffrey V. Lazarus, Nina Weis, Anne Øvrehus, Peer Brehm Christensen.

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
