## [Decision Letter · Decision Letter 0]

14 Jul 2020

PONE-D-20-12246

Hepatitis C prevalence in Denmark in 2016

- an updated estimate using multiple national registers

PLOS ONE

Dear Dr. Christensen,

Thank you for submitting your manuscript to PLOS ONE. After careful consideration, we feel that it has merit but does not fully meet PLOS ONE’s publication criteria as it currently stands. Therefore, we invite you to submit a revised version of the manuscript that addresses the points raised during the review process.

Your manuscript was reviewed by 2 experts in the field. Both identified many important problems in your submission. Please carefully review the attached comments and provide clear point-by-point responses.

We look forward to receiving your revised manuscript.

Kind regards,

Yury E Khudyakov, PhD

Academic Editor

PLOS ONE

Journal Requirements:

2. In the ethics statement in the manuscript and in the online submission form, please provide additional information about the participant information used in your retrospective study.

Specifically, please ensure that you have discussed whether all data were fully anonymized before you accessed them and/or whether the IRB or ethics committee waived the requirement for informed consent.

If patients provided informed written consent to have data from these registries/databases used in research, please include this information.

4.Thank you for stating the following in the Financial Disclosure section:

'The study was supported by an unrestricted grant from MSD Denmark

URL: https://www.msd.dk/home

The funders had no role in study design, data collection and analysis, decision to publish, or preparation of the manuscript.'

We note that you received funding from a commercial source: MSD Denmark

Within this Competing Interests Statement, please confirm that this does not alter your adherence to all PLOS ONE policies on sharing data and materials by including the following statement: "This does not alter our adherence to PLOS ONE policies on sharing data and materials.” (as detailed online in our guide for authors http://journals.plos.org/plosone/s/competing-interests).  If there are restrictions on sharing of data and/or materials, please state these.

Please note that we cannot proceed with consideration of your article until this information has been declared.

Reviewers' comments:

Reviewer's Responses to Questions

**Comments to the Author**

1. Is the manuscript technically sound, and do the data support the conclusions?

Reviewer #1: Partly

Reviewer #2: No

2. Has the statistical analysis been performed appropriately and rigorously? 

Reviewer #1: Yes

Reviewer #2: Yes

3. Have the authors made all data underlying the findings in their manuscript fully available?

Reviewer #1: Yes

Reviewer #2: Yes

4. Is the manuscript presented in an intelligible fashion and written in standard English?

Reviewer #1: Yes

Reviewer #2: Yes

5. Review Comments to the Author

Reviewer #1: The paper is of very poor value from the scientific point of view. The estimate of the proportion of subjects with undiagnosed CHC is affected by important biases: prisoners aren't representative of the general population, indeed findings lack of external validity; data from sera collected more than 20 years ago (1998-2000) are extremely out-dated.

Furthermore, the epidemiological data presented are obsolete because, starting from 2014, the epidemiological context has completely changed following the new direct-acting oral antiviral therapies (DAA) which, with a high sustained viral response (> 95%) and the consequent increase in treatments, have led to a faster decline of chronic hepatitis C prevalence globally.

HCV infection is mostly asymptomatic end random population-based surveys may provide accurate figures. Several national surveys have been performed in France, Spain, Italy, and U.S.A. over the last few years. None of them has been commented in the Discussion and reported in the References.

Reviewer #2: PONE-D-20-12246

The authors used a capture recapture method to estimate the population living with HCV in Denmark in the year 2016. The concept of this study is of value and the results will contribute to the literature. However, I have some comments regarding the analysis, and the content of the study.

I have below a few comments and suggestions that may improve the readability and understandability of the article.

Major comments:

1- In the method section, the first paragraph placement is confusing (line 82-line 86). I would suggest placing this paragraph after the data sources listing since this is highlighting the indicator used for the recapture stage.

2- In the Results section, the 2nd paragraph was confusing and hard to follow. The excluded 1,292 patients from the hospital register should be mentioned in the first paragraph where the authors are listing how they ended up with the number of patients included in the analysis (Line 190-198).

3- Also, I don’t understand how these patients were still counted since they didn’t meet the case definition mentioned in the methods section above: "1,292 who had either negative antiHCV or HCV-RNA results and/or had chronic hepatitis B (ICD10 code 18.1) with no signs of HCV coinfection according to the other source registers."

Minor comments:

1- Method section line 96, the word "included" is missing: " Since 1995, it also included all 97 hospital outpatient and emergency department visits

6. PLOS authors have the option to publish the peer review history of their article (what does this mean?). If published, this will include your full peer review and any attached files.

Reviewer #1: **Yes: **Filomena Morisco

Reviewer #2: No

---

## [Author Response · Author response to Decision Letter 0]

25 Jul 2020

Author response to queries in bold italic (please se uploaded file Response to Reviewers)

2. In the ethics statement in the manuscript and in the online submission form, please provide additional information about the participant information used in your retrospective study.

Specifically, please ensure that you have discussed whether all data were fully anonymized before you accessed them and/or whether the IRB or ethics committee waived the requirement for informed consent.

ANSWER: Data were fully anonymized prior to analysis. As a register-based study without contact to patients, informed consent from participants was not required, according to Danish law. This has been added to the methods section

If patients provided informed written consent to have data from these registries/databases used in research, please include this information. 

ANSWER: Patients registered in the clinical database (DANHEP) provided written consent to have data used in research. This has been specified in the methods section.

ANSWER: We have obtained accept from our data safety authority to publish the anonymized dataset. This is available as an Excel-file. (DOI: 10.5281/zenodo.3959476)

4.Thank you for stating the following in the Financial Disclosure section:

'The study was supported by an unrestricted grant from MSD Denmark

URL: https://www.msd.dk/home

The funders had no role in study design, data collection and analysis, decision to publish, or preparation of the manuscript.'

We note that you received funding from a commercial source: MSD Denmark

ANSWER: we have provided a specific statement of the funding role. 

Within this Competing Interests Statement, please confirm that this does not alter your adherence to all PLOS ONE policies on sharing data and materials by including the following statement: "This does not alter our adherence to PLOS ONE policies on sharing data and materials.” (as detailed online in our guide for authors http://journals.plos.org/plosone/s/competing-interests). If there are restrictions on sharing of data and/or materials, please state these.

Please note that we cannot proceed with consideration of your article until this information has been declared.

ANSWER: we have included the statement in the cover letter. 

Reviewers' comments:

Reviewer's Responses to Questions

Comments to the Author

1. Is the manuscript technically sound, and do the data support the conclusions?

Reviewer #1: Partly

Reviewer #2: No

2. Has the statistical analysis been performed appropriately and rigorously? 

Reviewer #1: Yes

Reviewer #2: Yes

3. Have the authors made all data underlying the findings in their manuscript fully available?

Reviewer #1: Yes

Reviewer #2: Yes

4. Is the manuscript presented in an intelligible fashion and written in standard English?

Reviewer #1: Yes

Reviewer #2: Yes

5. Review Comments to the Author

Reviewer #1: The paper is of very poor value from the scientific point of view. The estimate of the proportion of subjects with undiagnosed CHC is affected by important biases: prisoners aren't representative of the general population,

ANSWER As for the external validity, we agree that our samples are small and biased, but they are the only ones available in the country at the moment as no systematic HCV screening at population level is available. In addition, general population surveys also have drawbacks: most patients with chronic HCV are people with current or previous drug use and this population is usually not well captured in general population surveys (See e.g. the ECDC technical protocol for hepatitis C prevalence surveys in the general population published in March 2020. https://www.ecdc.europa.eu/en/publications-data/toolkit-support-generation-robust-estimates-hepatitis-c-prevalence). We agree that prison populations are not representative of the general population, but drug users may be tested here and we did not use the study to estimate the HCV prevalence but the undiagnosed fraction of chronic hepatitis C.

indeed findings lack of external validity; 

ANSWER We agree that it would have been of high scientific value to perform a systematic population based survey of HCV prevalence in Denmark. However, we have not had the resources to do this. As an example testing a randomly selected population of 5000 individuals would cost 80.000€ in test kits –not including the costs of contacting and collecting the samples- and with an estimated population prevalence of 0.2% such a study would likely only identify about 10 patients with chronic HCV. In contrast the total cost of our study was less than 25.000 €. With the reservations mentioned in the discussion we find it is a cheap and feasible method to produce a national HCV prevalence estimate for Denmark.

data from sera collected more than 20 years ago (1998-2000) are extremely out-dated: 

ANSWER: We find that the 3B samples are not outdated: There is no ongoing infection in this cohort and the few participants with HCV were infected in their youth in 60ies-80ies through drug use or nosocomial transmission prior to HCV screening introduced in the 90ies. They had not been identified in the registers since 2000 and therefor still represent the undiagnosed hepatitis C in the “pre-baby-boomer generation” that is still alive. 

Furthermore, the epidemiological data presented are obsolete because, starting from 2014, the epidemiological context has completely changed following the new direct-acting oral antiviral therapies (DAA) which, with a high sustained viral response (> 95%) and the consequent increase in treatments, have led to a faster decline of chronic hepatitis C prevalence globally.

ANSWER: Unfortunately our data is not obsolete; In Denmark restrictions for DAA use were not removed until Nov. 2018 and few patients were treated between 2016 and 2018. Our study shows that in Denmark there will be a greater impact on HCV prevalence by contacting and treating diagnosed patients than by screening for undiagnosed HCV patients. Also importantly, our data will provide the denominator to measure the current Danish efforts to reach the WHO targets of 90% diagnosed and 80% treated. 

HCV infection is mostly asymptomatic end random population-based surveys may provide accurate figures. Several national surveys have been performed in France, Spain, Italy, and U.S.A. over the last few years. None of them has been commented in the Discussion and reported in the references.

ANSWER: We are aware of the national population-based HCV prevalence surveys conducted in many countries -reviewed by e.g. Hofstraat et. al. 2017- and have added a reference to this in the discussion section. We have also added a short paragraph and reference to a recent HCV modelling study which found a similar proportion of diagnosed among people living with chronic HCV in Denmark to further support the data from the two external sources used in our study.

Reviewer #2: PONE-D-20-12246

The authors used a capture recapture method to estimate the population living with HCV in Denmark in the year 2016. The concept of this study is of value and the results will contribute to the literature. However, I have some comments regarding the analysis, and the content of the study.

I have below a few comments and suggestions that may improve the readability and understandability of the article.

Major comments:

1- In the method section, the first paragraph placement is confusing (line 82-line 86). I would suggest placing this paragraph after the data sources listing since this is highlighting the indicator used for the recapture stage.

ANSWER: We have mowed the paragraph as suggested.

2- In the Results section, the 2nd paragraph was confusing and hard to follow. The excluded 1,292 patients from the hospital register should be mentioned in the first paragraph where the authors are listing how they ended up with the number of patients included in the analysis (Line 190-198).

ANSWER: We have joint the two paragraphs and clarified why we don’t think the 1292 patients had hepatitis C. The bottom line is that the hospital register is not as reliable as the other registers we use, and we have removed patients who in the other registers had clear indication not to have chronic hepatitis C.

3- Also, I don’t understand how these patients were still counted since they didn’t meet the case definition mentioned in the methods section above: "1,292 who had either negative antiHCV or HCV-RNA results and/or had chronic hepatitis B (ICD10 code 18.1) with no signs of HCV coinfection according to the other source registers."

ANSWER: The 1.292 had the hepatitis C code in the hospital register (ICD code 18.2). However they had a later negative test for HCV in the laboratory register, or had no HCV test results and were instead positive for HBsAg or HBVDNA indicating chronic hepatitis B and suggesting a typing error in the hospital register where hepatitis B is coded as 18.0 or 18.1 We have clarified this in the text.

Minor comments:

1- Method section line 96, the word "included" is missing: " Since 1995, it also included all 97 hospital outpatient and emergency department visits

ANSWER: line 96 includes corrected to included”, and “it also all” changed to “it also included a

---

## [Decision Letter · Decision Letter 1]

12 Aug 2020

Hepatitis C prevalence in Denmark in 2016

- an updated estimate using multiple national registers

PONE-D-20-12246R1

Dear Dr. Christensen,

We’re pleased to inform you that your manuscript has been judged scientifically suitable for publication and will be formally accepted for publication once it meets all outstanding technical requirements.

Kind regards,

Yury E Khudyakov, PhD

Academic Editor

PLOS ONE

Additional Editor Comments (optional):

Reviewers' comments:

Reviewer's Responses to Questions

**Comments to the Author**

1. If the authors have adequately addressed your comments raised in a previous round of review and you feel that this manuscript is now acceptable for publication, you may indicate that here to bypass the “Comments to the Author” section, enter your conflict of interest statement in the “Confidential to Editor” section, and submit your "Accept" recommendation.

Reviewer #2: All comments have been addressed

2. Is the manuscript technically sound, and do the data support the conclusions?

Reviewer #2: Yes

3. Has the statistical analysis been performed appropriately and rigorously? 

Reviewer #2: Yes

4. Have the authors made all data underlying the findings in their manuscript fully available?

Reviewer #2: Yes

5. Is the manuscript presented in an intelligible fashion and written in standard English?

Reviewer #2: Yes

6. Review Comments to the Author

Reviewer #2: (No Response)

7. PLOS authors have the option to publish the peer review history of their article (what does this mean?). If published, this will include your full peer review and any attached files.

Reviewer #2: No

---

## [Editor Report · Acceptance letter]

26 Aug 2020

PONE-D-20-12246R1 

Hepatitis C prevalence in Denmark in 2016
- an updated estimate using multiple national registers 

Dear Dr. Christensen:

I'm pleased to inform you that your manuscript has been deemed suitable for publication in PLOS ONE. Congratulations! Your manuscript is now with our production department. 

Kind regards, 

on behalf of

Dr. Yury E Khudyakov 

Academic Editor

PLOS ONE